# Individual-specific functional connectivity improves prediction of Alzheimer's disease's symptoms in elderly people regardless of *APOE ε4* genotype

Lin Hua[1,2], Fei Gao[3], Xiaoluan Xia[1,2], Qiwei Guo[1,2], Yonghua Zhao [4], Shaohui Huang[5] & Zhen Yuan [1,2✉]

To date, reliable biomarkers remain unclear that could link functional connectivity to patients' symptoms for detecting and predicting the process from normal aging to Alzheimer's disease (AD) in elderly people with specific genotypes. To address this, individual-specific functional connectivity is constructed for elderly participants with/without *APOE ε4* allele. Then, we utilize recursive feature selection-based machine learning to reveal individual brain-behavior relationships and to predict the symptom transition in different genotypes. Our findings reveal that compared with conventional atlas-based functional connectivity, individual-specific functional connectivity exhibits higher classification and prediction performance from normal aging to AD in both *APOE ε4* groups, while no significant performance is detected when the data of two genotyping groups are combined. Furthermore, individual-specific between-network connectivity constitutes a major contributor to assessing cognitive symptoms. This study highlights the essential role of individual variation in cortical functional anatomy and the integration of brain and behavior in predicting individualized symptoms.

[1] Faculty of Health Sciences, University of Macau, Avenida da Universidade, Taipa, Macau SAR 999078, China. [2] Centre for Cognitive and Brain Sciences, University of Macau, Avenida da Universidade, Taipa, Macau SAR 999078, China. [3] Institute of Modern Languages and Linguistics, Fudan University, Shanghai 200433, China. [4] State Key Laboratory of Quality Research in Chinese Medicine, Institute of Chinese Medical Sciences, University of Macau, Avenida da Universidade, Taipa, Macau SAR 999078, China. [5] Institute of Biophysics, Chinese Academy of Sciences, Beijing 100101, China. ✉email: zhenyuan@um.edu.mo

Alzheimer's disease (AD) is one of the most prevalent neurodegenerative disorders, causing deficits in memory and executive function and difficulties in patients' daily life[1]. Accumulated evidence has demonstrated that AD progression is closely associated with specific genotypes[2,3] and the spectrum of AD clinical phenotypes[4,5]. Meanwhile, it is challenging to early detect and predict the process from normal aging (NA) to AD at the individual level, particularly for patients with different clinical phenotypes. Therefore, it is essential to inspect the brain-behavior relationship and identify the crucial biomarkers from NA to AD in elderly people with specific genotypes.

Interestingly, human *Apolipoprotein E* (*APOE*) genotype variants are one of the major genome-wide associated risk factors for AD in elderly people[6]. The *APOE* gene has three gene alleles (i.e., *APOE ε2*, *APOE ε3*, and *APOE ε4*), thus producing six genotypes (i.e., *ε2/ε2*, *ε2/ε3*, *ε2/ε4*, *ε3/ε3*, *ε3/ε4*, and *ε4/ε4*). In particular, existing studies illustrated that the presence of ε4 allele dose-dependently enhances the risk of AD, with *ε3/ε4* carriers and ε4/ε4 carriers showing 3.68-fold and 7-fold increased risks as compared to ε3 homozygotes e.g.,[7]. In addition to the high risk for AD, *APOE ε4* carriers are inclined to exhibit poorer scores in cognitive functions, episodic memory, executive function, and perceptual speed as compared to *APOE ε4* noncarriers[8,9]. Further, the brain networks in both *APOE* genotyping populations have been inspected, which indicated that both *APOE ε4* carriers and *APOE ε4* noncarriers manifest brain disconnection and decreased functional connectivity mainly in the default mode network (DMN)[10,11]. These studies suggested that *APOE ε4* carriers and *APOE ε4* noncarriers may present distinct neural pathways causing the differing stages from NA to AD. Therefore, it is essential to divide elderly participants into different *APOE* genotyping groups, so as to obtain higher performance for classifying and predicting the process from NA to AD.

Elderly participants might exhibit varying cognitive symptoms since they could be at different stages of cognitive decline or at different levels of disease progression. Therefore, identifying the brain biomarkers with sufficient accuracy to track particular symptoms at the individual level would fundamentally benefit the way of assessment and management in clinical practice. In particular, neuroimaging like structural and functional magnetic resonance imaging (MRI) has enabled us to make advances in identifying the potential biomarkers of AD. Likewise, brain network analysis has been performed to detect the macroscopic features that can distinguish well between various categories of dementia[12]. However, existing evidence has not yet been integrated into a set of regional interactions which can effectively track each elderly individual's present AD-related symptoms from NA to AD and assist in the evaluation of individuals with different genotypes.

In addition, biomarkers detected from the circuit anomalies among NA, mild cognitive impairment (MCI), and AD have been compromised by the inconsistent brain activation regions identified across individuals. Thus, stable parcellations of the cerebral cortex[13–15] and subcortical structures[16,17] are essential for the large-scale investigation into human brain architecture, which provides a cortical taxonomy for inspecting regional or network-level alterations in cortical functions associated with symptoms. However, the downside of these atlases is that they only offer the functional organization of the brain at the population level rather than the peculiarities of a particular individual. To date, a majority of neuroimaging studies still relied on group-level atlases for accessing individual-level functional data[10,12]. Although these atlas-level analyses were able to identify the relationships between brain connectivity and clinical demographic data[18,19], some nuanced yet critical information was still missing since the symptoms associated with functional networks are highly variable across individuals. For example, recent studies showed that some essential characteristics of brain networks might be missing by the use of atlas-based templates[14,20]. Consequently, these findings might dilute the brain-behavior correlations that are crucial for fully understanding the neural mechanism underlying the process from NA to AD when using the group-level atlas on individual participants.

In this study, an individual-specific strategy[21] that examined the individual differences in cortical functional architecture was proposed to improve the robustness and inter-individual reliability of functional connectivity analysis and to generate the relationship between the brain networks and participants' individualized symptoms. Our findings suggested that functional connectivity varied widely across individuals, particularly in the frontal and parietal cortices which were associated with higher-order cognitive functions. Next, to quantify the relationship between brain and clinical symptoms, we examined the individual-specific functional connectivity of a large cohort of *APOE ε4* carriers and noncarriers with varying degrees of cognitive symptoms, respectively. Specifically, machine learning was carried out to identify the connections that track multiple domains of cognitive symptoms [e.g., Mini-Mental State Examination (MMSE) and Immediate Recall Total Score (LIMM)] in different *APOE ε4* groups. Our results demonstrated that the use of individual-specific functional connectivity can detect effective biomarkers of clinical symptoms to facilitate early detection and prediction from NA to AD in both *APOE ε4* groups. Furthermore, we categorized whole-brain functional connections as within-network or between-network ones. We found that individual-specific between-network connectivity mostly contributed to assessing cognitive symptoms in both *APOE ε4* groups. Therefore, our study demonstrated the critical contribution of accounting for individual variation in cortical functional anatomy to tracking multiple clinical symptoms and genotypes, which could open an avenue for the diagnosis and prediction from NA to AD.

## Results

**Inter-subject variability in size, position, and functional connectivity of individuals' brain regions**. In light of an iteratively individual-specific functional network parcellation approach, a total of 235 elderly participants were initially destined into *APOE ε4* carriers ($N = 120$) and noncarriers ($N = 115$) (Table 1). Then, 18 cortical networks were mapped in each individual, and 116 discrete ROIs were derived from these networks. Individuals showed inter-individual variability in these functional ROIs (see Supplementary Fig.1 for example). For each participant, functional connectivity was calculated among these ROIs in order to investigate the relationship between the brain and behavior with specific genotypes (Fig. 1).

To examine whether the individual-specific approach carried high variation information across individuals and whether individual-specific connectivity conducted a higher correlation to individual differences than atlas-based connectivity, this study quantified the individual variability in individual-specific connectivity, atlas-based connectivity, and vertex-based connectivity across all 235 participants. The relationship between individual-specific connectivity and individual variability in size and position of the 116 ROIs was then evaluated. Individual variability in vertex-based connectivity showed that functional connectivity was highly variable across individuals, especially in the frontal and parietal cortices which were associated with higher-order cognitive functions (Fig. 2a). Additionally, vertex-based functional connectivity was significantly associated with the variability in connectivity strength among individual-specific ROIs ($r = 0.27$, $p = 0.003$; Fig. 2b, d) and atlas-based ROIs ($r = 0.20$, $p = 0.032$; Fig. 2c, e). Furthermore,

**Table 1 Demographic characteristics of all participants.**

| | | NA (N = 42) | MCI (N = 39) | AD (N = 39) | P-value | Total (N = 120) |
|---|---|---|---|---|---|---|
| *APOE ε4* carriers | Age, years | 69.80 (5.48) | 71.87 (6.18) | 73.18 (7.84) | 0.057 | 71.37 (6.52) |
| | Gender, male/female | 17/25 | 23/16 | 17/22 | 0.07 | 50/70 |
| | Education, years | 15.75 (2.76) | 15.64 (2.11) | 15.63 (3.09) | 0.228 | 15.71 (2.16) |
| | MMSE | 29.22 (0.84) | 27.65 (2.3) | 22.33 (4.3) | <0.001 | 26.84 (3.88) |
| | LIMM | 14.62 (2.95) | 9.49 (4.21) | 4.58 (2.98) | <0.001 | 10.28 (5.34) |
| | | **NA (N = 43)** | **MCI (N = 39)** | **AD (N = 33)** | **P-value** | **Total (N = 115)** |
| *APOE ε4* noncarriers | Age, years | 70.61 (5.79) | 73.02 (8.58) | 72 (8.02) | 0.275 | 71.67 (7.25) |
| | Gender, male/female | 19/24 | 22/17 | 14/19 | 0.094 | 55/60 |
| | Education, years | 16.29 (2.42) | 16.23 (2.61) | 16.17 (3.15) | 0.293 | 16.26 (2.39) |
| | MMSE | 29.23 (0.87) | 27.95 (1.62) | 21.4 (2.64) | <0.001 | 27.43 (3.24) |
| | LIMM | 15.11 (2.67) | 9.87 (3.37) | 3.7 (2.72) | <0.001 | 10.35 (5.14) |

Note: Data is presented as mean ± standard deviations (SD). *NA* normal aging, *MCI* mild cognitive impairment, *AD* Alzheimer's disease.

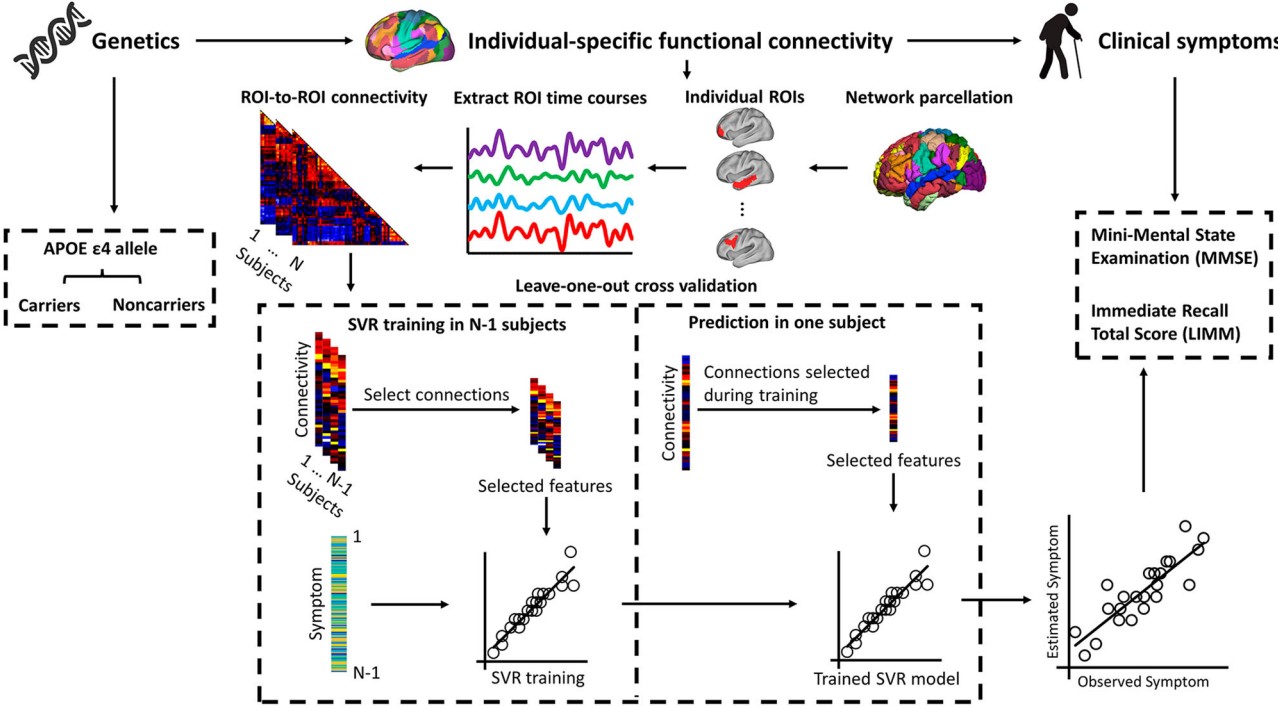

**Fig. 1 Procedure of estimating symptom scores in elderly people with/without *APOE ε4* allele using functional connectivity among individually-specified ROIs.** Participants were initially classified as *APOE ε4* carriers (N = 120) and noncarriers (N = 115) depending on whether they carried at least one *APOE ε4* allele or not, respectively. Then, based on the individual-level cortical network parcellation approach, we identified 116 homologous functional ROIs in each individual participant. The rs-fMRI signals in each ROI were then extracted and functional connectivity among these ROIs was computed, resulting in a 116 × 116 connectivity matrix for each participant. SVR model was trained to estimate each participant's symptom scores based on ROI-to-ROI connectivity. To reduce the dimensionality of the input data, only a subset of connections that showed significant correlations with the symptom scores in the training dataset were selected as the relevant features to train the SVR. Data from *N*-1 participants were used to train the model and then the resulting model was applied to the data of the remaining participants to estimate the individual's symptoms. This procedure was repeated *N* times to predict the symptom scores of all participants. The correlation between the estimated and observed behavioral scores was then evaluated.

individual-specific connectivity showed a higher correlation with vertex-based connectivity than individual variability in atlas-based connectivity. Then, the individual variability in individual-specific connectivity was significantly associated with both variability in size ($r = 0.66$, $p < 0.001$; Fig. 3a, c) and variability in position ($r = 0.27$, $p = 0.004$; Fig. 3b, d). Therefore, these results could justify that individual-specific connectivity was more related to different aspects of individual variability than atlas-based connectivity, and the significant results in the current study corresponding to the individual-specific approach were caused by the individual variability across participants.

**Distinguishing among NA, MCI, and AD in the individual-specific/atlas-based functional connectivity.** To examine whether individual-specific functional connectivity can effectively distinguish the elderly participants among NA, MCI, and AD, we developed classifiers that predicted the clinical subgroups based on individual-specific and atlas-based functional connectivity across different *APOE* genotypes. Consistent with previous studies[22,23], the classification of MCI from NA or AD was less accurate than separating NA from AD (Table 2). Furthermore, compared with classification performance using atlas-based functional connectivity, individual-specific functional connectivity performed

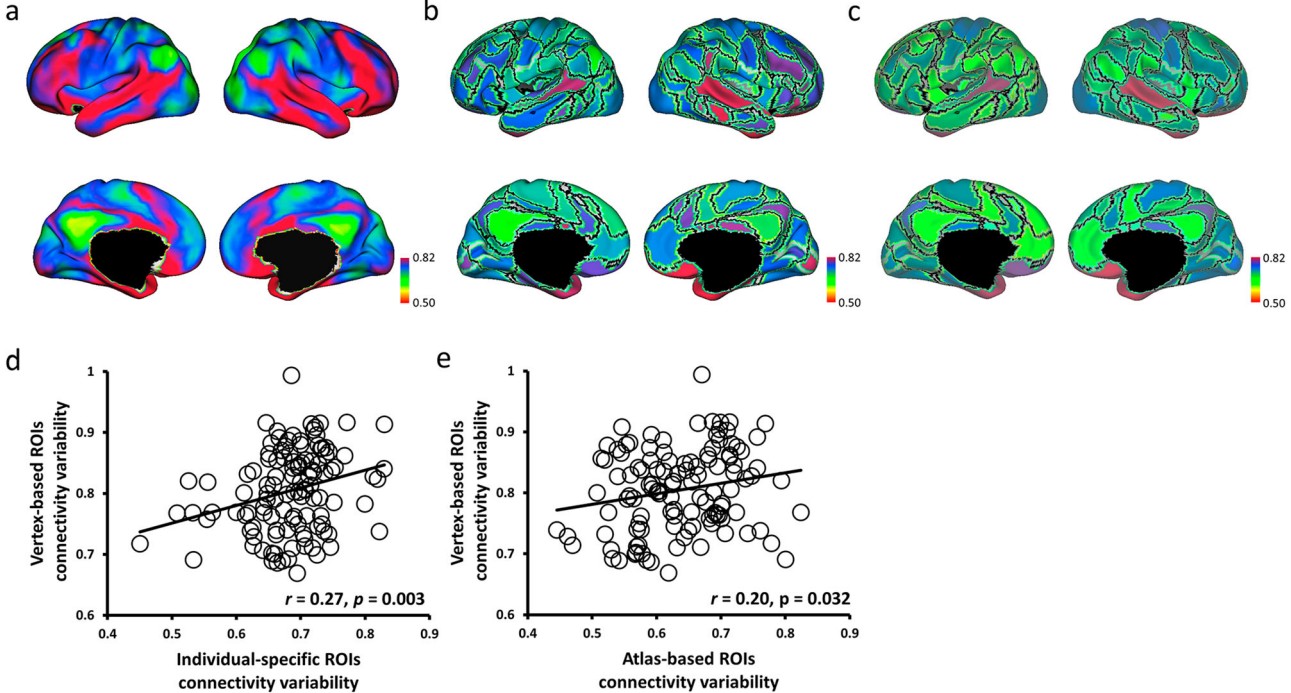

**Fig. 2 Individual-specific ROI connectivity was more correlated with vertex-based ROI connectivity than with atlas-based ROI connectivity. a** Individual variability in resting state functional connectivity was calculated at each vertex across participants (*N* = 235). Frontal and parietal cortices showed stronger individual variability than other cortices. **b, c** Individual-specific ROI connectivity and atlas-based ROI connectivity were quantified at 116 ROIs across participants (*N* = 235). **d, e** Both individual variability in individual-specific ROI (*r* = 0.27, *p* = 0.003) and atlas-based ROI (*r* = 0.20, *p* = 0.032) connectivity showed significant association with individual variability in vertex-based ROI connectivity (*N* = 116 ROIs).

better in the evaluation matrix of accuracy (ACC), specificity (SPE), sensitivity (SEN), and area under the receiver operating characteristic curve (AUC), especially in *APOE ε4* carriers (Table 2).

**Individual-specific functional connectome tracks MMSE symptoms**. To determine whether individual-specific functional connectivity in different *APOE* genotyping carriers could track cognitive status, SVR model was trained to estimate the MMSE scores from each participant carrying or not carrying *APOE ε4* allele. For the individuals with *APOE ε4* allele, we found that a collection of functional connections among individual-specific ROIs were able to reliably predict MMSE symptom ratings. Meanwhile, the estimated and observed MMSE scores showed a modest yet statistically significant correlation (*r* = 0.41, *p* = 0.025, Fig. 4a). Connections that exerted the greatest prediction on the MMSE symptom mainly employed the FPN and DMN (Fig. 4b and Supplementary Fig. 4a). In contrast, MMSE scores estimated by atlas-based functional connectivity identified in Yeo's group-level atlas[13] were not significantly correlated with the observed MMSE symptoms (*r* = 0.33, *p* = 0.087, Fig. 4c). For the individuals without *APOE ε4* allele, neither individual-specific ROIs (*r* = 0.23, *p* = 0.264) nor atlas-based ROIs (*r* = 0.21, *p* = 0.252) were able to estimate MMSE scores.

Focusing on the symptom-related connections, we then examined whether the same connections defined by atlas-based ROIs would be less correlated with symptom scores. In other words, we tested if the connectivity features were already impaired by the atlas-based functional connectivity before the prediction model was applied. We found that the symptom-related connections (Fig. 4b) were less correlated with symptom scores (Supplementary Fig. 3) indeed when defined by atlas-based ROIs. This finding suggested that the atlas-based ROIs obscured

the symptom-related connections, thus further impeding symptom prediction.

**Individual-specific functional connectome tracks LIMM symptoms**. To examine the specificity and precision of the current approach in estimating multiple symptoms, we next tested whether individual-specific functional connectivity can track LIMM symptom scores in the same group of individuals with or without *APOE ε4* allele. For the group of *APOE ε4* carriers, significantly positive correlation (*r* = 0.30, *p* = 0.048, Fig. 5a) was again obtained between the estimated and observed LIMM symptom scores. Nevertheless, functional connectivity among the atlas-based ROIs cannot predict the LIMM scores (*r* = 0.23, *p* = 0.13, Fig. 5c). Connections most contributing to LIMM symptom prediction mainly involved the FPN, DMN, and sensorimotor (MOT) (Fig. 5b and Supplementary Fig. 4b). For the *APOE ε4* noncarrier group, functional connectivity among individual-specific ROIs was able to predict LIMM symptom scores (*r* = 0.44, *p* = 0.012, Fig. 6a). Although atlas-based functional connectivity also showed a significant correlation between the estimated and observed LIMM scores, the correlation was relatively weaker as compared to the individual-based functional connectivity (*r* = 0.38, *p* = 0.041, Fig. 6c). Connections most contributing to LIMM symptom prediction mainly involved the FPN, ATN, DMN, and MOT (Fig. 6b and Supplementary Fig. 4c). Additionally, decreased correlation with symptom scores was also found with the symptom-related connections (Figs. 5b and 6b) defined by atlas-based functional connectivity (Supplementary Fig. 3).

**Between-network connectivity is a major contributor to the estimation of symptoms**. By examining the functional connections that were predictive of MMSE and LIMM symptoms

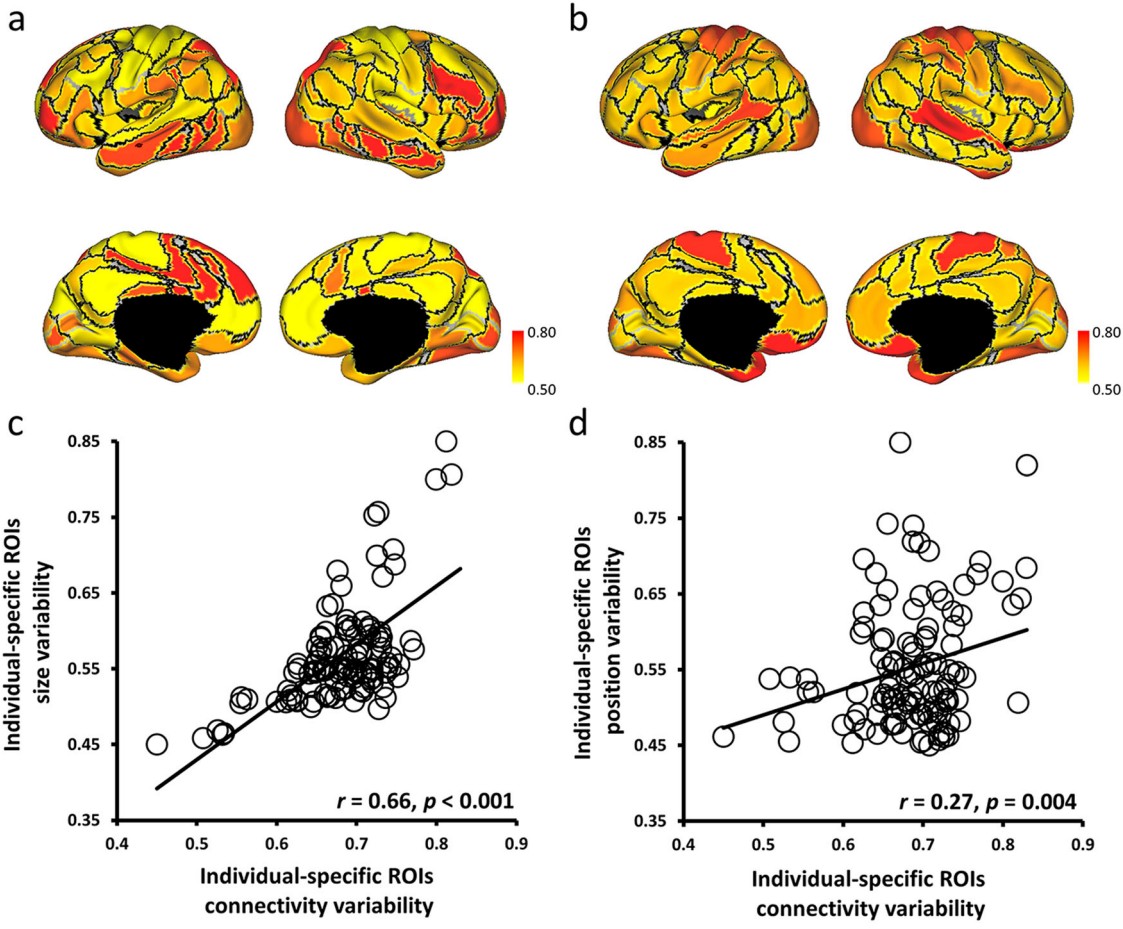

**Fig. 3 Individual variability in individual-specific ROI connectivity was significantly associated with individual variability in the size and position of the functional regions. a, c** Individual variability in ROI size was calculated for each of the 116 ROIs, and the size variability showed a significant correlation ($r = 0.66$, $p < 0.001$) with the variability in individual-specific ROI connectivity. **b, d** Individual variability in ROI position was also quantified for each of the 116 ROIs, and the position variability showed a moderate correlation ($r = 0.27$, $p = 0.004$) with the variability in individual-specific ROI connectivity.

**Table 2 Classification performance between clinical subgroups in individual-specific/atlas-based functional connectivity.**

| | | SVM Classification | AUC | ACC | SEN | SPE |
|---|---|---|---|---|---|---|
| APOE ε4 carriers (N = 120) | Individual-specific FC | NA vs. MCI | 0.94 | 0.88 | 0.92 | 0.85 |
| | | MCI vs. AD | 0.92 | 0.87 | 0.73 | 0.94 |
| | | NA vs. AD | 0.94 | 0.90 | 0.92 | 0.85 |
| | Atlas-based FC | NA vs. MCI | 0.90 | 0.82 | 0.94 | 0.65 |
| | | MCI vs. AD | 0.86 | 0.75 | 0.70 | 0.80 |
| | | NA vs. AD | 0.92 | 0.88 | 0.90 | 0.80 |
| APOE ε4 noncarriers (N = 115) | Individual-specific FC | NA vs. MCI | 0.92 | 0.84 | 0.98 | 0.64 |
| | | MCI vs. AD | 0.89 | 0.81 | 0.50 | 1.00 |
| | | NA vs. AD | 0.96 | 0.89 | 0.94 | 0.82 |
| | Atlas-based FC | NA vs. MCI | 0.90 | 0.83 | 0.96 | 0.64 |
| | | MCI vs. AD | 0.89 | 0.81 | 0.50 | 1.00 |
| | | NA vs. AD | 0.96 | 0.87 | 0.94 | 0.78 |

*Individual-specific FC* Individual-specific functional connectivity, *Atlas-based FC* Atlas-based functional connectivity.

(e.g., Figs. 4b, 5b, and 6b), we found that the majority of them were connections between functional networks rather than those within the same network. The strength of between-network connectivity showed a decrease of 7.25% on average when the ROIs were individual-specific compared to atlas-based ($p < 0.01$

for 17 of the 18 networks, paired t-test, Bonferroni corrected for 18 comparisons, Supplementary Fig. 2). Therefore, the reduced connectivity could result in more accurate symptom estimations, suggesting that between-network connectivity could be quantified more precisely when functional regions were localized in individuals.

Grouping the connections into the 7 canonical functional networks, the connections that contributed to MMSE symptom estimate model were mostly between-network connections including the FPN and DMN in *APOE ε4* carriers (Fig. 7a). Furthermore, the connections that contributed to LIMM symptom estimation model were mostly between-network connections including the FPN, DMN, and MOT in both *APOE ε4* carriers (Fig. 7b) and *APOE ε4* noncarriers (Fig. 7c). Specifically, ATN showed a higher contribution ranking in *APOE ε4* noncarriers than in *APOE ε4* carriers.

**Estimations of symptom dimensions perform better within different *APOE* genotyping groups.** While a large body of imaging and neuropathology studies tended to suggest some distinct aspects of pathophysiology in elderly people with different *APOE* genotypes[3,24], it is still unclear whether different functional connectivity could result in distinct neuropsychological representations in elderly people with differing *APOE* genotypes. To this end, we conducted an SVM group separation of *APOE ε4* carriers and *APOE ε4* noncarriers in differing clinical groups, respectively. Furthermore, we aggregated all 235

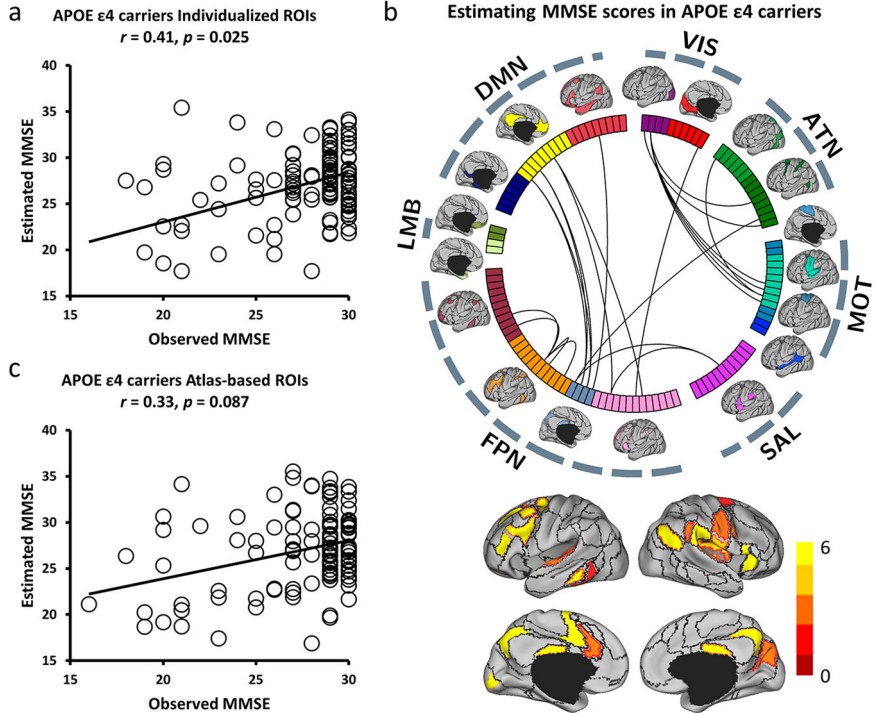

**Fig. 4 Functional connectivity among the individual-specific ROIs can better predict MMSE symptoms than that among the atlas-based ROIs in *APOE ε4* carriers from NA to AD. a** The scatterplot demonstrates the correlation (Pearson's correlation, *r* = 0.41, *p* = 0.025) between the MMSE scores predicted by the connectivity among the individual-specific ROIs and the actually observed scores in *APOE ε4* carriers from NA to AD (*N* = 120). Each circle indicates a subject. Correlation significance was determined by using 1000 permutations. **b** 116 ROIs derived from the 18 networks are presented by the colored rectangles under the corresponding brain networks. Twenty-two connections, which are above the 90th percentile of absolute weight in MMSE symptom estimation among *APOE ε4* carriers (*N* = 120), are specified by the black lines. Group-level maps of the 18 functional networks are shown on the cortical surface, respectively. ROIs involved in these predictive connections are plotted and color-coded with differing weights on the cortical surface below. **c** A similar analysis is performed using 116 ROIs defined in an atlas-based template. Functional connectivity among the atlas-based ROIs is not able to predict the MMSE scores in the group of *APOE ε4* carriers (*N* = 120; *r* = 0.33, *p* = 0.087). Correlation significance was determined by using 1000 permutations.

individuals across both *APOE ε4* carriers and *APOE ε4* non-carriers, and then trained SVR models to predict MMSE and LIMM symptoms on this merged dataset.

The SVM group separation of *APOE ε4* carriers and *APOE ε4* noncarriers using 10-fold cross-validation revealed an AUC of 0.98 (ACC = 0.89, SEN = 0.75, and SPE = 0.97) in AD groups, an AUC of 0.91 (ACC = 0.80, SEN = 0.80, and SPE = 0.81) in MCI groups, and an AUC of 0.86 (ACC = 0.78, SEN = 0.88, and SPE = 0.68) in NA groups (Fig. 8a). Furthermore, even though the aggregated analysis included the largest number of participants, individual-specific functional connectivity in this cross-genotype cohort cannot perform MMSE symptom estimation above the chance level (*r* = 0.18, *p* = 0.139, Fig. 8b). Similar results were also found in LIMM symptom estimation. Individual-specific functional connectivity failed to yield LIMM estimation, which was not significantly correlated with the observed LIMM symptom rating (*r* = 0.23, *p* = 0.108, Fig. 8c).

## Discussion
Drawing on the individual-specific functional network parcellation and machine learning approaches, the current study aimed at identifying the neuroimaging signatures in different *APOE* genotyping groups that could track cognitive symptoms from NA to AD. Our methods would enable data-driven estimations of the specific cortical connections which allow for the most accurate prediction of clinical statuses and symptom scores in the elderly population with or without *APOE ε4* allele. Specifically, the

current results provide reliable readouts for testing how changes in a particular model affect the model's ability to predict the clinical subgroups and estimate a specific set of symptoms, which further establishes the relationship between brain functional connectivity and behavioral symptoms.

In line with previous studies[25–27], individual variability results demonstrated that functional connectivity was highly variable across individuals, especially in the higher-order cognitive areas. The individual-specific approach used in the current study can more precisely depict individual differences than the atlas-based approach. Furthermore, the comparison results indicated that compared to performing atlas-based functional connectivity, analysis carried out on individual-specific functional regions could promote the prediction of patients' clinical statuses and the correspondence between functional connectivity and multiple symptom scores in differing *APOE* genotypes. This pattern was in line with the evidence from the healthy population[28] and psychotic illness patients[29], which showed that individual-specific functional connectivity manifested much greater accuracy to assess fluid intelligence and psychotic symptoms than using the atlas-based template.

With regard to the connections most contributing to symptom estimation, previous studies reported that *APOE* gene-related symptoms could alter the resting-state functional connectivity among the frontal, temporal, and DMN regions[24,30,31]. Yet, convergent evidence was unable to be obtained from NA to AD population[32,33], even though this is thought to be a gradual process from NA to MCI and finally AD[34]. In the current work,

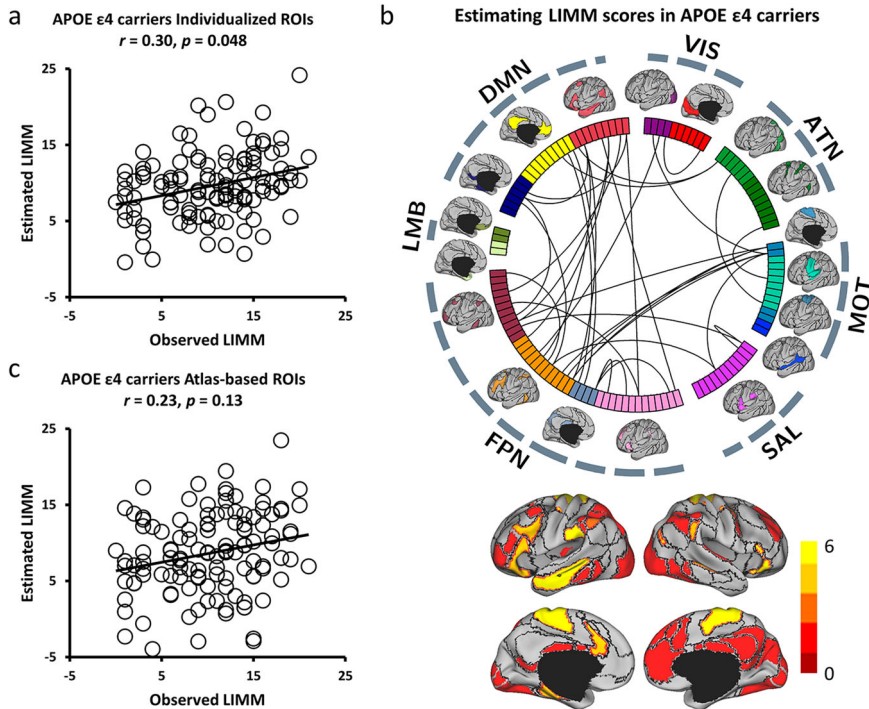

**Fig. 5 Functional connectivity among the individual-specific ROIs can better predict LIMM symptoms than that among the atlas-based ROIs in *APOE ε4* carriers from NA to AD. a** The scatterplot demonstrates the correlation (Pearson's correlation, $r = 0.30$, $p = 0.048$, 1000 permutation test) between the LIMM scores predicted by the connectivity among the individual-specific ROIs and the actually observed scores ($N = 120$). **b** Thirty-four connections, which are above the 90th percentile of absolute weight in LIMM symptom estimation among *APOE ε4* carriers ($N = 120$), are denoted by the black lines. ROIs involved in these predictive connections are plotted and color-coded with differing weights on the cortical surface below. **c** Functional connectivity among the atlas-based ROIs is not able to well predict the LIMM scores in the group of *APOE ε4* carriers ($N = 120$; $r = 0.23$, $p = 0.13$, 1000 permutation test).

we observed the most consistent results in *APOE ε4* carriers that FPN and DMN connections contributed to MMSE symptom estimation model, while the FPN, DMN, and MOT connections contributed to LIMM symptom estimation model in *APOE ε4* carriers and noncarriers. Moreover, compared to the results in *APOE ε4* carriers, the contribution ranking of ATN was higher in *APOE ε4* noncarriers, thus suggesting that ATN might constitute a critical region corresponding to symptoms in *APOE ε4* noncarriers. This finding is compatible with several proposals examining the abnormal circuit in MCI or AD patients relative to healthy participants[31,35].

Another noteworthy finding from the current research is that, in every symptom we tested, between-network connection in both *APOE ε4* carriers and noncarriers implicated a critical role in predicting the severity of symptoms. Although the absolute values of between-network connectivity were significantly reduced after functional alignment, they still performed a more accurate prediction of symptoms. While exploring interactions across functional networks from NA to AD participants, previous studies primarily focused on the within-network variation as a key source of the illness-related signal. In particular, the hyper- or hypo-connectivity within DMN was reported to be associated with the extent of cognitive decline[33]. In this study, however, less relationship between within-network connectivity and clinical symptoms was identified, especially in the elderly participants with *APOE ε4* allele. Since within-network variance may only reflect symptom-related general pathology of the disorder, the results of variation in between-network connectivity may represent the identification of symptom-related specific biomarkers. This finding is in line with recent studies suggesting that changes in between-network connectivity may signify neuropathology changes in diseases[29,36]. Importantly, our findings indicate that

accounting for individual differences in functional network boundaries is critical, as mislocalization of networks might obscure the actually low between-network correlations and further the identification of brain-behavior associations. To our knowledge, our study is among the first to combine individual-specific functional connectivity and machine learning approaches to examine the relationships among brain and behavioral symptoms in elderly people with specific genotypes from NA to AD. Therefore, the analysis framework of the current study can be extended to the investigations of brain-behavior associations in both healthy and clinical populations in the future.

Finally, some caveats need to be noted regarding the present study. Our study did not subdivide the groups of *APOE ε4* carriers or noncarriers into NA, MCI, and AD since symptom ratings might be low or invariant among individuals in the same subgroup. Meanwhile, subdividing groups might reduce the effective numbers of sampling and bias the symptom estimation models. Moreover, we investigated the neuroimaging signatures of their correlation with symptom scores from *APOE ε4* carriers and noncarriers, respectively, instead of differentiating the two groups. Finally, although feature selection was conducted using significant connectivity related to symptoms, the use of LOOCV still might increase the chance of overfitting. Future investigations could address these issues by subgrouping the participants in an increased sample size and performing various machine-learning models to verify the estimated findings.

In summary, the current study found that the connectivity between individual-specific functional areas in elderly participants with different *APOE* genotypes was capable of predicting the clinical subgroups from NA to AD and yielding moderate-to-strong estimation levels across many primary categories of AD-related symptomatology. Notably, without accommodating

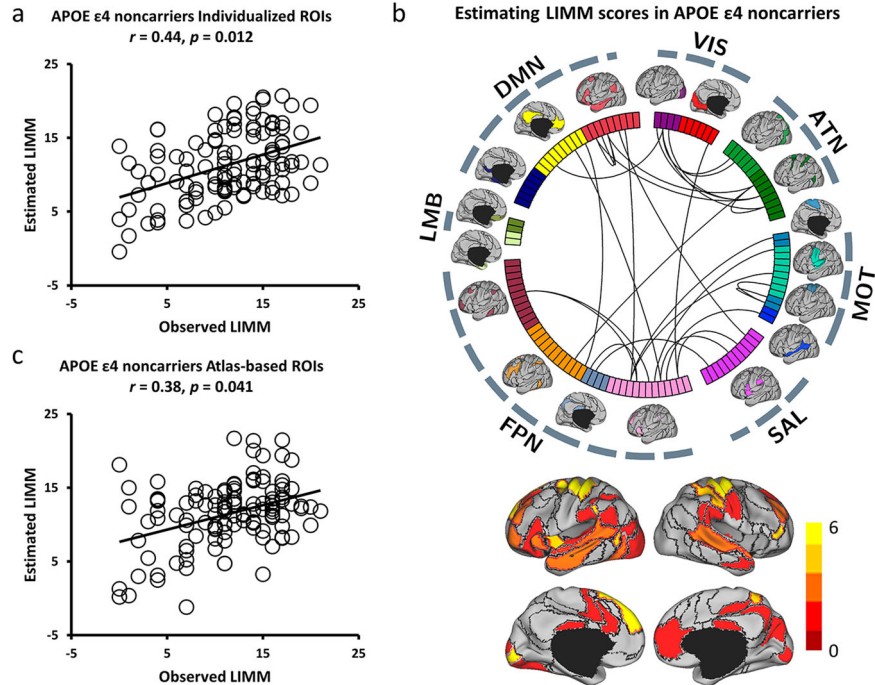

**Fig. 6 Functional connectivity among the individual-specific ROIs can better predict LIMM symptoms than that among the atlas-based ROIs in *APOE ε4* noncarriers from NA to AD. a** The scatterplot demonstrates the correlation (Pearson's correlation, $r = 0.44$, $p = 0.012$, 1000 permutations) between the LIMM scores predicted by the connectivity among the individual-specific ROIs and the observed scores ($N = 115$). **b** Twenty-nine connections, which are above the 90th percentile of absolute weight in LIMM symptom estimation among *APOE ε4* noncarriers ($N = 115$), are plotted by the black lines. ROIs involved in these predictive connections are plotted and color-coded with differing weights on the cortical surface below. **c** The correlation between the predicted and observed LIMM scores was weaker when connectivity was estimated using the atlas-based ROIs among these participants ($N = 115$; $r = 0.38$, $p = 0.041$, 1,000 permutations).

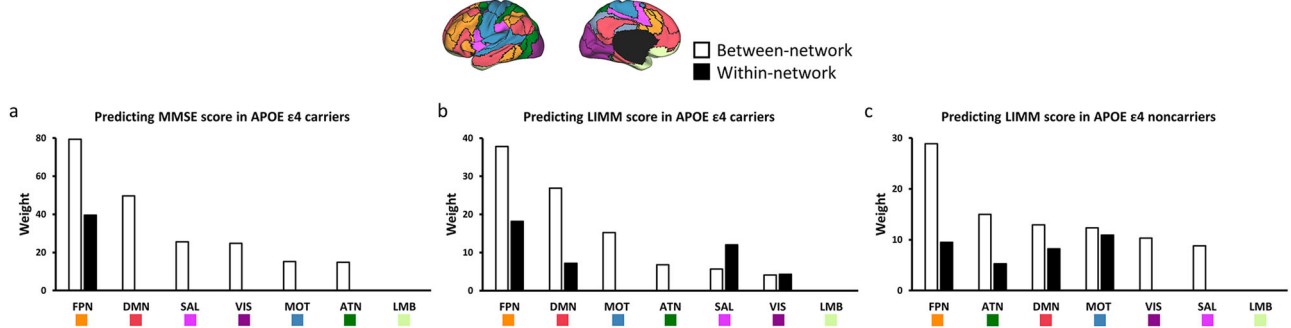

**Fig. 7 Between-network connectivity plays an essential role in predicting MMSE symptoms in *APOE ε4* carriers and LIMM symptoms in *APOE ε4* carriers and *APOE ε4* noncarriers. a** The functional connections most predicted of the MMSE scores in *APOE ε4* carriers ($N = 120$) are grouped according to the 7 canonical networks. Connections contributing to the symptom prediction are mainly between-network connections (white bars). These between-network connections mainly involve the FPN and DMN. **b** The prediction of LIMM scores in *APOE ε4* carriers ($N = 120$) is mainly driven by between-network (white bars) involving the FPN, DMN, and MOT. **c** The prediction of LIMM scores in *APOE ε4* noncarriers ($N = 115$) is mainly driven by the between-network involving the FPN, ATN, DMN, and MOT. Connections within the VIS and SAL also contributed to the prediction.

individual differences in cortical functional architecture, conventional atlas-based functional connectivity was shown to be ineffective in predicting these clinical statuses and symptoms in almost all cases. Furthermore, between-network variation explained far more symptom-specific severity than in atlas-defined models, especially for elderly people with *APOE ε4* allele. Our findings underscored the importance of accounting for individual variation in cortical functional anatomy in neurodegenerative research, which can be extended to the healthy population as well. Moreover, precision medicine, which aims to provide personalized treatment strategies by considering individualized disease heterogeneity, could benefit from using individual-specific functional connectivity models to tailor treatments to specific needs and improve patient care. Therefore, our work highlighted the meaningful relationship between brain connectivity and symptoms, which can serve clinical utility in diagnosis, prognosis, and treatment for elderly people from potential to identified disorders.

## Methods
**Participants**. Participants were retrieved from the phase 2 and phase 3 datasets from the Alzheimer's Disease Neuroimaging Initiative (ADNI; https://adni.loni.usc.edu/) in light of the availability of T1-weighted and resting-state functional MRI, *APOE* genotypes, and symptom severity assessment including MMSE and LIMM.

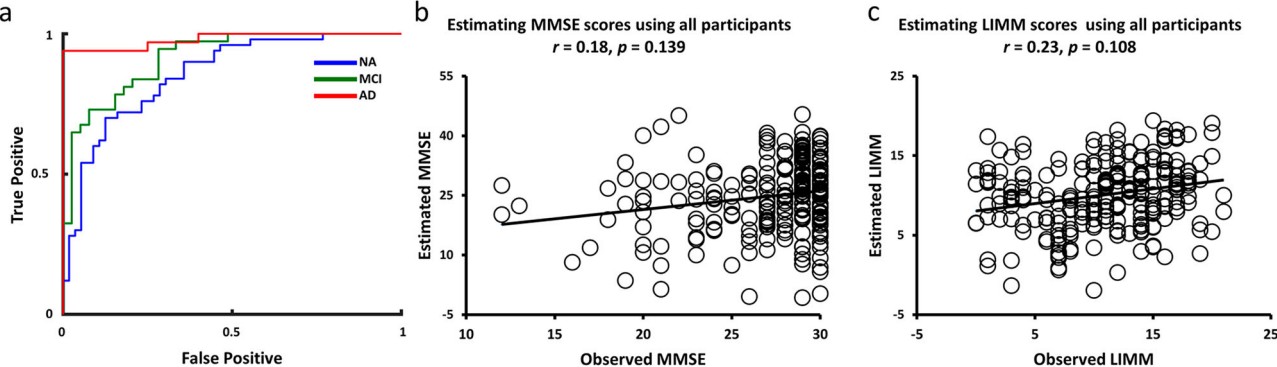

**Fig. 8 Prediction of clinical groups and symptoms across *APOE* genotyping groups. a** The AUCs for *APOE* ε4 carriers versus *APOE* ε4 noncarriers for the 5-fold cross-validation in NA (42 *APOE* ε4 carriers versus 43 *APOE* ε4 noncarriers), MCI (39 *APOE* ε4 carriers versus 39 *APOE* ε4 noncarriers), and AD groups (39 *APOE* ε4 carriers versus 33 *APOE* ε4 noncarriers), respectively. **b** Functional connectivity was not able to estimate the MMSE scores in the elderly participants across two *APOE* genotypes (*N* = 235). **c** Functional connectivity was also not able to estimate the LIMM scores in the cross-genetic cohort (*N* = 235).

According to *APOE* genotypes, participants were classified into two groups: 1) *APOE* ε4 carriers' group with at least one *APOE* ε4 allele (genotype ε3/ε4 and ε4/ε4), 2) *APOE* ε4 noncarriers' group with genotype ε3/ε3. Previous studies have verified that *APOE* ε2 allele could contaminate the dose effect of the *APOE* ε4 allele and result in different neuropathology[37,38]. Thus, individuals with ε2 allele (i.e., ε2/ε2, ε2/ε4, and ε2/ε3) were excluded due to the possible protective effect. In addition, neuroimaging data of participants with excessive head motions (see the third step of neuroimaging data preprocessing), severe artifacts, partial brain coverage, histories of obvious head trauma, and alcohol/drug abuse were also excluded for further analysis. Further, 235 elderly participants (120 *APOE* ε4 carriers and 115 *APOE* ε4 noncarriers) were selected for the present study (Table 1). Ethical approval was approved by the institutional review boards of ADNI participating institutions, and written informed consent was obtained from all participants.

**APOE *genotyping, neuropsychological assessment, and neuroimaging data acquisition*.** As previously described in ref. [39], all participants' *APOE* genotypes were screened by using DNA extracted from peripheral blood cells. The cells were collected in 10 ml EDTA plastic tubes and transported to the University of Pennsylvania AD Biofluid Bank Laboratory by overnight delivery at room temperature. More detailed information can be found in the ADNI Procedures Manual (http://adni.loni.usc.edu/methods/documents/).

A battery of neuropsychological assessments was completed by each participant in ADNI database. The present study focused on the results of MMSE and LIMM, which have been well-validated and widely used as reliable tools for assessing cognitive impairment[40]. Importantly, the two assessments have been reported to be associated with brain functional connectivity[41,42]. Table 1 presents the demographic and neuropsychological performances of all participants.

Both structural and functional MRI data of the baseline were included in this study. Functional MRI (fMRI) data were selected based on the following parameters: TR = 3000 ms; TE = 30 ms; flip angle = 90°; number of slices = 48; slice thickness = 3.4 mm. The first section (200-time points, 10 min) of fMRI data was extracted as each participant's resting-state data. Detailed MRI scanner protocols for structural and functional sequences are available online (http://adni.loni.usc.edu/methods/documents/mri-protocols/).

**Neuroimaging data preprocessing.** Resting-state fMRI (rs-fMRI) data were processed in terms of previous protocols[29,43] that include the following steps: 1) removal of the first four frames; 2) slice timing correction with the FSL package[44] (http://www.fmrib.ox.ac.uk/fsl/); 3) rigid-body correction for head motion using FSL. Framewise displacement (FD) and root-mean-square of voxel-wise differentiated signal (DVARS) were then estimated using fsl_motion_outliers implemented in FSL. Volumes with FD > 0.2 mm or DVARS > 50 were marked as outliers (censored frames). One frame before and two frames after these outlier volumes were also flagged as censored frames, together with those lasting fewer than five contiguous volumes. Volumes with more than half labeled censored frames were removed; 4) linear regression of multiple nuisance regressors consisting of a vector of ones and linear trend, six motion correction parameters, averaged white matter signal, averaged ventricular signal, and temporal derivatives of the six motion correction parameters, averaged white matter signal, and averaged ventricular signal; 5) interpolation of censored frames with Lomb-Scargle periodogram; 6) band-pass filtering (0.009–0.08 Hz).

Structural MRI data were processed using the FreeSurfer 7.1.1 package (http://surfer.nmr.mgh.harvard.edu). The structural and functional images were aligned using boundary-based registration[45]. In a single interpolation, rs-fMRI data were aligned to a spherical coordinate system by sampling from the cortical ribbon.

fMRI data of each individual was initially registered to the FreeSurfer surface template, which had 40962 vertices in each hemisphere. A 6 mm full-width half-maximum (FWHM) smoothing kernel was then applied to the fMRI data in the surface space. The smoothed data were then down-sampled to a mesh of 2562 vertices in each hemisphere using the mri_surf2surf function in FreeSurfer.

**Identifying functional ROIs in individuals.** The functional regions of interest (ROIs) for individual participants were localized by using the previous methods[21,28,29] with the following procedures:

Step 1. Individual participants' cortical functional networks were mapped by using the iterative parcellation method[21]. The algorithm was initially guided by the population-level functional network atlas constructed from 1000 healthy participants. As the impact of the atlas on each individual brain parcellation was not identical for every participant or each brain area, the atlas was flexibly altered depending on the known distribution of inter-individual variability and the signal-to-noise ratio (SNR) distribution in a given participant. As the iteration progressed, the effect of population-based data diminished, enabling the final map to be totally driven by the data of each individual participant. More detailed information on the population-level functional network atlas and the iterative functional parcellation algorithm can be found in Wang, et al.[21].

Step 2. Using a clustering algorithm (mri_surfcluster in FreeSurfer software), the cortical networks of individual participants obtained from Step 1 were divided into a number of discrete patches. Each cortical network on the surface was spatially smoothed using a Gaussian kernel function (sigma = 1 mm) to reduce the influence of noise and matching costs. Only the template matching procedure (as explained below) would be impacted by the smoothing. The original unsmoothed area was kept for further analysis after a homologous ROI was detected.

Step 3. Individual participants' discrete patches were matched to the 116 cortical ROIs generated from the population-level atlas that guided the search for a participant's networks. The template matching procedure was performed for each cortical network as follows: 1) If an individual-level patch was overlapped (more than 20 vertices) with a single ROI in the population-level network, the patch was labeled as the same ROI in the atlas; 2) If a single individual-level patch was overlapped with numerous ROIs from a single network, the patch was divided into smaller patches. Vertices that overlapped with the population-level ROIs were labeled together with these ROIs, producing the centers of numerous smaller patches. According to the geodesic distance on the brain surface, the remaining vertices in the original patch were allocated to the closest ROIs; 3) If a patch was not overlapped with any population-level ROI and the shortest distance between the patch and the ROI was less than a specific threshold, the patch was allocated to the ROI closest to it. The mean distance between any two vertices in the closest ROI was used as the specific threshold in the procedure; otherwise, the patch was labeled as "unrecognized".

**Estimating within-network and between-network functional connectivity.** Individual-specific functional connectivity was computed using Pearson correlation, resulting in a 116 × 116 connectivity matrix for each participant. For the comparison between individual-specific and atlas-based functional connectivity, Yeo's group-level atlas[13] was used to form a similar 116 × 116 connectivity matrix for each participant. Then, 116 ROIs based on both individual and atlas levels were divided into 18 networks[21]. Functional connections were classified as within-network or between-network depending on whether they connected two ROIs in the same or different networks, respectively. Finally, each participant's within-network and between-network connection values were calculated. The within-network connectivity was quantified by the averaged connectivity values of all ROI

pairs within the same network, while the between-network connectivity was measured by the averaged connectivity values of all ROI pairs that involved one ROI within the specific network and the other ROIs outside this network.

**Estimating individual variability in size, position, and vertex-wise and ROI-based connectivity of the functional regions**. To further examine the individual variability in the size and position of a functional region, as well as the vertex-wise and ROI-based connectivity were estimated using the similar strategy described in Li et al.[28]. The size of a functional region was estimated by the number of vertices that fell within this functional region. Then, the size variability for each ROI was quantified by the standard deviation of the size of functional regions across participants. The position of a functional region was determined by the coordinates of the center mass in this functional region. Then, the averaged geodesic distance between the ROI centers across participants was used to estimate the position variability for each ROI. Vertex-wise functional connectivity was denoted by the connectivity between vertices according to fsLR_32 k surface mesh (59412 vertices). Furthermore, individual variability in ROI-based functional connectivity was computed as the strength of individual-specific ROI-to-ROI functional connectivity and atlas-based ROI-to-ROI functional connectivity across participants.

**Predicting individual symptoms using individual-specific/atlas functional connectivity**. Based on the connection between ROIs, a nonlinear support vector machine (SVM) with a radial basis function (RBF) kernel from the LIBSVM toolbox (https://www.csie.ntu.edu.tw/~cjlin/libsvm/) was trained to assess the prediction performance of clinical subgroups among NA, MCI, and AD in different genotyping groups, and genotyping groups between *APOE ε4* carriers and noncarriers in different clinical subgroups. Covariates, including age and gender, were initially regressed from the connection features prior to feature selection. Then, to avoid the over-fitting issue due to the 6670 connection features (symmetric matrix with $116 \times 116$) and remove the duplicate information, statistical significances between each pair of two groups (NA vs. MCI, NA vs. AD, and MCI vs. AD in different genotyping groups, and *APOE ε4* carriers vs. *APOE ε4* noncarriers in different clinical subgroups) were conducted by a two-sample two-sided *t*-test. The 5-fold cross-validation method was used to train the model on the de-confounded data from 80% of participants and test the model on the remaining 20% of participants. Finally, ACC, SPE, SEN, and AUC were calculated to evaluate the performance of the classification model.

To predict each participant's symptom severity ratings based on the connection between ROIs, a SVM for regression algorithm (L2-regularized L2-loss SVR model) from LIBLINEAR toolbox (https://www.csie.ntu.edu.tw/~cjlin/liblinear/) was trained. The leave-one-out cross-validation (LOOCV) method was utilized, in which data from N-1 participants were used to train the model. And then the model was applied to the data of the remaining participants to assess the severity of the participant's symptoms. Before feature selection, covariates including age and gender were regressed from the features and the symptom severity ratings. The regressing weights were applied to the left-out dataset. To avoid over-fitting and remove duplicate information, features that exhibited strong associations with symptom ratings in the training dataset were chosen to train the model in each LOOCV[46]. Then, the de-confounded features from the testing data were fed into the trained model to calculate the predicted symptom ratings. To assess the symptom scores of all individuals, the procedure was repeated N times ($N = 120$ or $N = 115$). Finally, the estimated and observed symptom ratings were compared to reveal the correlation.

**Determining the contribution of each ROI and network in symptom estimation**. The weight of a connection in the SVR model was used to quantify its contribution. In the present study, brain connections above the 90th percentile of absolute weight in each symptom estimation were considered as the most reliable connections. Specifically, the SVR model in the training data generated a weight coefficient for each feature of each LOOCV fold. The weights of each connection were then averaged over all LOOCV folds to determine their contribution to symptom estimation. If a connection was not chosen as a feature in one-fold, its contribution in this fold was set to zero. A specific ROI's contribution was estimated by summing up the contributions of all connections affecting that ROI.

Between-network connections were separated from within-network connections to quantify the contribution of each functional network to symptom estimation (identical to the method of estimating within-network and between-network functional connectivity mentioned previously), which then grouped the predictive connections into the 7 canonical functional networks. The weight of each network in symptom estimation was calculated by adding the weights of the predicted connections of the involved network.

**Statistics and reproducibility**. Demographic characteristics were compared between NA, MCI, and AD in *APOE ε4* carriers and *APOE ε4* noncarriers using ANOVA for normally distributed and Kruskal-Wallis for non-normally distributed variables. Statistical significance was defined as $p < 0.05$ after Bonferroni correction.

For clinical subgroup classifications in SVM model, a significance criterion ($p = 0.001$) was used for feature selection between each pair of two groups. To predict each participant's symptom severity ratings in SVR model, significant

features associated with symptom ratings achieving the significant criteria ($p = 0.001$) after Bonferroni correction were chosen to train and test the SVR model, resulting in a small amount of characteristics. Furthermore, the significance of the correlation was assessed using permutation testing (1000 permutations), which randomly reshuffled the observed symptom among the participants. The *p*-value was estimated by calculating the percentage of the correlation value of permutation data higher than the correlation value of real data. The *p*-value was corrected for multiple comparisons by using the Bonferroni method.

**Visualization**. All imaging results were mapped onto the inflated PALS cortical surface and visualized by using CARET software (http://brainvis.wustl.edu/wiki/index.php/Caret:Download). Circos (http://circos.ca/) was used to construct the connectograms depicting connections that contributed to symptom estimation (e.g., Fig. 4b). Connections that contributed to symptom estimation were further split into those positively or negatively linked with the symptoms, which were elaborated in Supplementary Fig. 4.

**Reporting summary**. Further information on research design is available in the Nature Portfolio Reporting Summary linked to this article.

## Data availability
The data that support the findings of this study are publicly available from the ADNI dataset (https://adni.loni.usc.edu/) upon registration and compliance with the ADNI data use policy (https://ida.loni.usc.edu/collaboration/access/appLicense.jsp). Source data used to generate figures can be found in Supplementary Data 1–5.

## Code availability
All methods used open-source software, and all links to the relevant software are included in Methods (URLs). Code used in the analyses described in this paper is available at https://github.com/LinHuaUM/IndivCode.git.

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

## Acknowledgements

Data used in the preparation of this article were obtained from the Alzheimer's Disease Neuroimaging Initiative (ADNI) database (adni.loni.usc.edu). As such, the investigators within the ADNI contributed to the design and implementation of ADNI and/or provided data but did not participate in the analysis or writing of this report. A complete listing of ADNI investigators can be found at: http://adni.loni.usc.edu/wp-content/uploads/how_to_apply/ADNI_Acknow-ledgement_List.pdf Data collection and sharing for this project was funded by the Alzheimer's Disease Neuroimaging Initiative (ADNI) (National Institutes of Health Grant U01 AG024904) and DOD ADNI (Department of Defense award number W81XWH-12-2-0012). ADNI is funded by the National Institute on Aging, the National Institute of Biomedical Imaging and Bioengineering, and through generous contributions from the following: AbbVie, Alzheimer's Association; Alzheimer's Drug Discovery Foundation; Araclon Biotech; BioClinica, Inc.; Biogen; Bristol-Myers Squibb Company; Cere-Spir, Inc.; Cogstate; Eisai Inc.; Elan Pharmaceuticals, Inc.; Eli Lilly and Company; EuroImmun; F. Hoffmann-La Roche Ltd and its affiliated company Genentech, Inc.; Fujirebio; GE Healthcare; IXICO Ltd.; Janssen Alzheimer Immunotherapy Research & Development, LLC.; Johnson & Johnson Pharmaceutical Research & Development LLC.; Lumosity; Lundbeck; Merck & Co., Inc.; Meso Scale Diagnostics, LLC.; NeuroRx Research; Neurotrack Technologies; Novartis Pharmaceuticals Corporation; Pfizer Inc.; Piramal Imaging; Servier; Takeda Pharmaceutical Company; and Transition Therapeutics. The Canadian Institutes of Health Research is providing funds to support ADNI clinical sites in Canada. Private sector contributions are facilitated by the Foundation for the National Institutes of Health (www.fnih.org). The grantee organization is the Northern California Institute for Research and Education, and the study is coordinated by the Alzheimer's Therapeutic Research Institute at the University of Southern California. ADNI data are disseminated by the Laboratory for Neuro Imaging at the University of Southern California. This work was also supported by the University of Macau (MYRG 2020-00067-FHS and MYRG 2019-00082-FHS), Macao Science and Technology Development Fund (FDCT 0048/2021/AGJ and FDCT 0020/2019/AMJ), and Higher Education Fund of Macao SAR Government (CP-UMAC-2020-01).

## Author contributions

L.H.: conceptualization, data curation, methodology, software, investigation, formal analysis, writing—original draft, writing—review and editing, visualization. F.G.: conceptualization, methodology, resources, writing—review and editing. Xl.X.: data curation, formal analysis, writing—review and editing. QW.G.: methodology, formal analysis, writing—review and editing. Yh.Z.: conceptualization, methodology, writing—review and editing. Sh.H.: conceptualization, writing—review and editing. Z.Y.: conceptualization, methodology, funding acquisition, project administration, supervision, writing—review and editing.

## Competing interests

The authors declare no competing interests.
