## [Peer Review File · Communications Biology]

Reviewers' comments:

Reviewer #1 (Remarks to the Author):

In this manuscript, the authors examined individual functional architecture to predict the clinical symptoms of Alzheimer's disease in APOE e4 carriers/noncarriers. They found that compared with the traditional atlas-based approach, individual functional connectivity features showed an improved performance for tracking the clinical symptoms, and they demonstrated that individualized between-network connectivity constituted a major contributor for accessing cognitive symptoms.

It is an important topic to investigate the influence of individual brain network in brain diseases with the novel approach. There were a few issues that should be addressed by authors and these issues are listed below:

Major issues:

1. The authors combined data to explore the distinct aspects of pathophysiology in APOE e4 carriers and noncarriers. It does not provide direct evidence of the difference between the two genotype groups. I would recommend using SVM to classify these two groups to increase the reliability of these conclusions. The results may show the separation between the two genotype groups.
2. In the discussion section, the authors should emphasize the concept of an individualized model capable of stratifying world real practice patients.

Minor issues:

1. Typos: Page 9. Last line. "a small amounts of characteristics" should be "a small amount of characteristics".
2. Figure 2A-C. Color bars should use the same range.

Reviewer #2 (Remarks to the Author):

Biomarkers for AD diagnosis and/or symptom progression are vital. Although the research field has been extensively explored, there is very limited progress in the identification of these markers. The current study by Hua and colleagues attempted to use individual-specific functional connectivity (FC, based on functional MRI), rather than population FC as previous studies used, to differentiate normal aging from AD. They claim that individual-specific FC was sufficient to separate NA from AD but failed to distinguish the different APOE genotypes. Although data presented in the manuscript appear to be interesting, several limitations weaken the significance of this study.

Major concerns:

1. The authors argue that they could "predict" symptom progression based on correlation coefficient of Observed MMSE and Estimated MMSE, in which the later was predicted from SVM training by using Observed MMSE. Generally speaking, the correlation coefficient is not an index for "prediction", but rather "variance explained".
The authors should further test whether individual-specific FC could be used to predict or classify patient into different subgroup, like NA, MCI or AD.
2. In the abstract, the authors claim that "no significant performance was detected when the data of two APOE genotyping groups were combined for the estimation models." I did not find evidence or data in the study supporting this statement. The authors should provide explicit results that support this claim.

Minor concerns:

1. Panel labels (A, B, C, ...) in Figures 4-8 are missing.

Response to Reviewers

COMMSBIO-23-0536A: Individual-specific functional connectivity improves Alzheimer's symptom prediction in elderly people with/without *APOE* $\epsilon 4$ allele

We thank the editors for considering our manuscript for publication in *Communications Biology*. We appreciate the reviewers' constructive comments for our manuscript.

To comply with the formatting guidelines and policy of *Communications Biology*, we revised the title and abstract of our manuscript in the revision. We further rewrote the final paragraph of introduction, which now includes the brief summary of the major results and conclusions, and added "Statistics and Reproducibility" in methods. To address the Reviewers' comments, we provided methodological details on predicting the clinical statuses (NA, MCI, and AD) of elderly people in different *APOE* $\epsilon 4$ groups and classifying the *APOE* $\epsilon 4$ genotypes of elderly people in different clinical subgroups. Moreover, we included further results on combining data analysis of two *APOE* $\epsilon 4$ groups. Additionally, we emphasized the concept of an individualized model capable of stratifying real-world practical patients in the last paragraph of discussion.

We hope this revised manuscript has addressed all the concerns, and look forward to hearing from you.

Response to Review 1:

Major Comments

Comment 1: *"The authors combined data to explore the distinct aspects of pathophysiology in *APOE* $\epsilon 4$ carriers and noncarriers. It does not provide direct evidence of the difference between the two genotype groups. I would recommend using SVM to classify these two groups to increase the reliability of these conclusions. The results may show the separation between the two genotype groups."*

Response: We would like to thank the reviewer for this insightful comment and took the suggestion accordingly. SVM group separations between *APOE* $\epsilon 4$ carriers and *APOE* $\epsilon 4$ noncarriers were conducted in different clinical subgroups (NA, MCI, and

AD), respectively. Specifically, the SVM classification between *APOE* $\epsilon 4$ carriers and *APOE* $\epsilon 4$ noncarriers in different clinical subgroups showed high performance with an AUC of 0.98 (ACC = 0.89, SEN = 0.75, and SPE = 0.97) in AD groups, an AUC of 0.91 (ACC = 0.80, SEN = 0.80, and SPE = 0.81) in MCI groups, and an AUC of 0.86 (ACC = 0.78, SEN = 0.88, and SPE = 0.68) in NA groups (Figure 8A).

These findings further confirmed the dissociative functional connectivity alterations between *APOE* $\epsilon 4$ carriers and *APOE* $\epsilon 4$ noncarriers in all clinical subgroups. We have provided the methodological details of SVM classification in the section of *Methods* (Line 14-27, Page 23) and analysis results in the section of “*Estimations of symptom dimensions perform better within different APOE genotyping groups*” (Line 9-12, Page 16).

Figure 8. Prediction of clinical groups and symptoms across *APOE* genotyping groups. (A) The AUCs for *APOE* $\epsilon 4$ carriers versus *APOE* $\epsilon 4$ noncarriers for the 5-fold cross-validation in NA, MCI, and AD groups, respectively. (B) Functional connectivity was not able to estimate the MMSE scores in the elderly participants across two *APOE* genotypes. (C) Functional connectivity was not able to estimate the LIMM scores in the cross-genetic cohort.

Comment 2: “*In the discussion section, the authors should emphasize the concept of an individualized model capable of stratifying world real practice patients.*”

Response: Thanks so much for pointing out this issue. As suggested, in the revision, we provided further clarification on the role of individualized model in precision medicine and highlighted the significance of our work in the field of clinical research (Line 9-16, Page 19).

Minor Comments

Comment 1: *“Typos: Page 9. Last line. “a small amounts of characteristics” should be “a small amount of characteristics.”*

Response: We thank the Reviewer for pointing this out and fixed it accordingly (Line 5, Page 25).

Comment 2: *“Figure 2A-C. Color bars should use the same range.”*

Response: As suggested, the new colored bars with the same range were plotted in the new Figure 2 (Page 8).

Response to Review 2:

Major Comments

Comment 1: *“The authors argue that they could “predict” symptom progression based on correlation coefficient of Observed MMSE and Estimated MMSE, in which the later was predicted from SVM training by using Observed MMSE. Generally speaking, the correlation coefficient is not an index for “prediction”, but rather “variance explained”.*

The authors should further test whether individual-specific FC could be used to predict or classify patient into different subgroup, like NA, MCI or AD.”

Response: We thank the Reviewer for the constructed comments on this important issue. In the revision, we provided methodological details on predicting the clinical statuses (NA, MCI, and AD) of elderly people in different *APOE ε4* groups using individual-specific/atlas-based functional connectivity in the section of *Method* (Line 14-27, Page 23). The methodological information included the SVM model we chose, feature selection, the 5-fold cross-validation method of training and testing the model, and model evaluation.

In light of these methods, we found that the classification of MCI from NA or AD was less accurate than separating NA from AD (Table 2). Furthermore, compared with classification performance using atlas-based functional connectivity, individual-

specific functional connectivity performed better in the evaluation matrix of accuracy (ACC), specificity (SPE), sensitivity (SEN), and area under the receiver operating characteristic curve (AUC), especially in *APOE* $\epsilon 4$ carriers (Table 2; Line 9-17, Page 9 and Line 1-7, Page 10).

Therefore, our findings provided further evidence that individual-specific FC showed improved prediction performance to classify participants with/without *APOE* $\epsilon 4$ allele into different clinical subgroups.

Table 2 Classification performance between clinical subgroups in individual-specific/atlas-based functional connectivity

		SVM Classification	AUC	ACC	SEN	SPE
APOE $\epsilon 4$ carriers	Individual-specific FC	NA vs. MCI	0.94	0.88	0.92	0.85
		MCI vs. AD	0.92	0.87	0.73	0.94
		NA vs. AD	0.94	0.90	0.92	0.85
	Atlas-based FC	NA vs. MCI	0.90	0.82	0.94	0.65
		MCI vs. AD	0.86	0.75	0.70	0.80
		NA vs. AD	0.92	0.88	0.90	0.80
APOE $\epsilon 4$ noncarriers	Individual-specific FC	NA vs. MCI	0.92	0.84	0.98	0.64
		MCI vs. AD	0.89	0.81	0.50	1.00
		NA vs. AD	0.96	0.89	0.94	0.82
	Atlas-based FC	NA vs. MCI	0.90	0.83	0.96	0.64
		MCI vs. AD	0.89	0.81	0.50	1.00
		NA vs. AD	0.96	0.87	0.94	0.78

Note: Individual-specific FC: Individual-specific functional connectivity; Atlas-based FC: Atlas-based functional connectivity.

Comment 2: “In the abstract, the authors claim that “no significant performance was detected when the data of two *APOE* genotyping groups were combined for the estimation models.” I did not find evidence or data in the study supporting this statement. The authors should provide explicit results that support this claim.”

Response: We thank the Reviewer for bringing up this point and apologize for not making it clear enough. To examine whether elderly people with differing *APOE* genotypes could result in the altered functional connectivity, we performed classification for *APOE* $\epsilon 4$ carriers and *APOE* $\epsilon 4$ noncarriers in different clinical subgroups and combined the data of two *APOE* genotyping groups for estimating clinical scores. In the revision, we added the analysis of group separation between

APOE $\epsilon 4$ carriers and *APOE* $\epsilon 4$ noncarriers in different clinical subgroups. The SVM classification results showed good performance with an AUC of 0.98 (ACC = 0.89, SEN = 0.75, and SPE = 0.97) in AD groups, an AUC of 0.91 (ACC = 0.80, SEN = 0.80, and SPE = 0.81) in MCI groups, and an AUC of 0.86 (ACC = 0.78, SEN = 0.88, and SPE = 0.68) in NA groups (Figure 8A).

Furthermore, we pooled the data from the two *APOE* genotyping groups and tracked clinical symptoms using SVR model. The results revealed that even though the aggregated analysis included the largest number of participants, individual-specific functional connectivity in this cross-genotype cohort were unable to estimate MMSE symptom above the chance level ($r = 0.18$, $p = 0.139$, Figure 8B). Similar results were also obtained in LIMM symptom estimation. Individual-specific functional connectivity failed to yield LIMM estimation which was not significantly correlated with the observed LIMM symptom rating ($r = 0.23$, $p = 0.108$, Figure 8C).

Our findings demonstrated the dissociative functional connectivity alterations between *APOE* $\epsilon 4$ carriers and *APOE* $\epsilon 4$ noncarriers in all clinical subgroups. We have rewritten this part and provided detailed results in the section of “*Estimations of symptom dimensions perform better within different APOE genotyping groups*” (Line 1-12, Page 16).

Figure 8. Prediction of clinical groups and symptoms across *APOE* genotyping groups. (A) The AUCs for *APOE* $\epsilon 4$ carriers versus *APOE* $\epsilon 4$ noncarriers for the 5-fold cross-validation in NA, MCI, and AD groups, respectively. (B) Functional connectivity was not able to estimate the MMSE scores in the elderly participants across two *APOE* genotypes. (C) Functional connectivity was not able to estimate the LIMM scores in the cross-genetic cohort.

Minor Comments

Comment 1: “Panel labels (A, B, C, ...) in Figures 4-8 are missing.”

Response: We appreciate the Reviewer for pointing out that. We have now added these panel labels in the updated Figures 4-8.

REVIEWERS' COMMENTS:

Reviewer #1 (Remarks to the Author):

This revision has well addressed my previous comments. I have no new comments. I would like to suggest to publish it as it is.

Reviewer #2 (Remarks to the Author):

The authors have fully answered my questions and concerns. I believe the manuscript is ready to be accepted, I have no further concerns.

Response to Reviewers

COMMSBIO-23-0536A: Individual-specific functional connectivity improves Alzheimer's symptom prediction in elderly people with/without *APOE* $\epsilon 4$ allele

We thank the editors for considering our manuscript for publication in *Communications Biology*. We appreciate the reviewers' constructive comments for our manuscript.

To comply with the formatting guidelines and policy of *Communications Biology*, we revised the title and abstract of our manuscript in the revision. We further rewrote the final paragraph of introduction, which now includes the brief summary of the major results and conclusions, and added "Statistics and Reproducibility" in methods. To address the Reviewers' comments, we provided methodological details on predicting the clinical statuses (NA, MCI, and AD) of elderly people in different *APOE* $\epsilon 4$ groups and classifying the *APOE* $\epsilon 4$ genotypes of elderly people in different clinical subgroups. Moreover, we included further results on combining data analysis of two *APOE* $\epsilon 4$ groups. Additionally, we emphasized the concept of an individualized model capable of stratifying real-world practical patients in the last paragraph of discussion.

We hope this revised manuscript has addressed all the concerns, and look forward to hearing from you.

Response to Review 1:

Major Comments

Comment 1: *"The authors combined data to explore the distinct aspects of pathophysiology in *APOE* $\epsilon 4$ carriers and noncarriers. It does not provide direct evidence of the difference between the two genotype groups. I would recommend using SVM to classify these two groups to increase the reliability of these conclusions. The results may show the separation between the two genotype groups."*

Response: We would like to thank the reviewer for this insightful comment and took the suggestion accordingly. SVM group separations between *APOE* $\epsilon 4$ carriers and *APOE* $\epsilon 4$ noncarriers were conducted in different clinical subgroups (NA, MCI, and

AD), respectively. Specifically, the SVM classification between *APOE* $\epsilon 4$ carriers and *APOE* $\epsilon 4$ noncarriers in different clinical subgroups showed high performance with an AUC of 0.98 (ACC = 0.89, SEN = 0.75, and SPE = 0.97) in AD groups, an AUC of 0.91 (ACC = 0.80, SEN = 0.80, and SPE = 0.81) in MCI groups, and an AUC of 0.86 (ACC = 0.78, SEN = 0.88, and SPE = 0.68) in NA groups (Figure 8A).

These findings further confirmed the dissociative functional connectivity alterations between *APOE* $\epsilon 4$ carriers and *APOE* $\epsilon 4$ noncarriers in all clinical subgroups. We have provided the methodological details of SVM classification in the section of *Methods* (Line 14-27, Page 23) and analysis results in the section of “*Estimations of symptom dimensions perform better within different APOE genotyping groups*” (Line 9-12, Page 16).

Figure 8. Prediction of clinical groups and symptoms across *APOE* genotyping groups. (A) The AUCs for *APOE* $\epsilon 4$ carriers versus *APOE* $\epsilon 4$ noncarriers for the 5-fold cross-validation in NA, MCI, and AD groups, respectively. (B) Functional connectivity was not able to estimate the MMSE scores in the elderly participants across two *APOE* genotypes. (C) Functional connectivity was not able to estimate the LIMM scores in the cross-genetic cohort.

Comment 2: “*In the discussion section, the authors should emphasize the concept of an individualized model capable of stratifying world real practice patients.*”

Response: Thanks so much for pointing out this issue. As suggested, in the revision, we provided further clarification on the role of individualized model in precision medicine and highlighted the significance of our work in the field of clinical research (Line 9-16, Page 19).

Minor Comments

Comment 1: *“Typos: Page 9. Last line. “a small amounts of characteristics” should be “a small amount of characteristics.”*

Response: We thank the Reviewer for pointing this out and fixed it accordingly (Line 5, Page 25).

Comment 2: *“Figure 2A-C. Color bars should use the same range.”*

Response: As suggested, the new colored bars with the same range were plotted in the new Figure 2 (Page 8).

Response to Review 2:

Major Comments

Comment 1: *“The authors argue that they could “predict” symptom progression based on correlation coefficient of Observed MMSE and Estimated MMSE, in which the later was predicted from SVM training by using Observed MMSE. Generally speaking, the correlation coefficient is not an index for “prediction”, but rather “variance explained”.*

The authors should further test whether individual-specific FC could be used to predict or classify patient into different subgroup, like NA, MCI or AD.”

Response: We thank the Reviewer for the constructed comments on this important issue. In the revision, we provided methodological details on predicting the clinical statuses (NA, MCI, and AD) of elderly people in different *APOE ε4* groups using individual-specific/atlas-based functional connectivity in the section of *Method* (Line 14-27, Page 23). The methodological information included the SVM model we chose, feature selection, the 5-fold cross-validation method of training and testing the model, and model evaluation.

In light of these methods, we found that the classification of MCI from NA or AD was less accurate than separating NA from AD (Table 2). Furthermore, compared with classification performance using atlas-based functional connectivity, individual-

specific functional connectivity performed better in the evaluation matrix of accuracy (ACC), specificity (SPE), sensitivity (SEN), and area under the receiver operating characteristic curve (AUC), especially in *APOE* $\epsilon 4$ carriers (Table 2; Line 9-17, Page 9 and Line 1-7, Page 10).

Therefore, our findings provided further evidence that individual-specific FC showed improved prediction performance to classify participants with/without *APOE* $\epsilon 4$ allele into different clinical subgroups.

Table 2 Classification performance between clinical subgroups in individual-specific/atlas-based functional connectivity

		SVM Classification	AUC	ACC	SEN	SPE
APOE $\epsilon 4$ carriers	Individual-specific FC	NA vs. MCI	0.94	0.88	0.92	0.85
		MCI vs. AD	0.92	0.87	0.73	0.94
		NA vs. AD	0.94	0.90	0.92	0.85
	Atlas-based FC	NA vs. MCI	0.90	0.82	0.94	0.65
		MCI vs. AD	0.86	0.75	0.70	0.80
		NA vs. AD	0.92	0.88	0.90	0.80
APOE $\epsilon 4$ noncarriers	Individual-specific FC	NA vs. MCI	0.92	0.84	0.98	0.64
		MCI vs. AD	0.89	0.81	0.50	1.00
		NA vs. AD	0.96	0.89	0.94	0.82
	Atlas-based FC	NA vs. MCI	0.90	0.83	0.96	0.64
		MCI vs. AD	0.89	0.81	0.50	1.00
		NA vs. AD	0.96	0.87	0.94	0.78

Note: Individual-specific FC: Individual-specific functional connectivity; Atlas-based FC: Atlas-based functional connectivity.

Comment 2: *“In the abstract, the authors claim that “no significant performance was detected when the data of two APOE genotyping groups were combined for the estimation models.” I did not find evidence or data in the study supporting this statement. The authors should provide explicit results that support this claim.”*

Response: We thank the Reviewer for bringing up this point and apologize for not making it clear enough. To examine whether elderly people with differing *APOE* genotypes could result in the altered functional connectivity, we performed classification for *APOE* $\epsilon 4$ carriers and *APOE* $\epsilon 4$ noncarriers in different clinical subgroups and combined the data of two *APOE* genotyping groups for estimating clinical scores. In the revision, we added the analysis of group separation between

APOE $\epsilon 4$ carriers and *APOE* $\epsilon 4$ noncarriers in different clinical subgroups. The SVM classification results showed good performance with an AUC of 0.98 (ACC = 0.89, SEN = 0.75, and SPE = 0.97) in AD groups, an AUC of 0.91 (ACC = 0.80, SEN = 0.80, and SPE = 0.81) in MCI groups, and an AUC of 0.86 (ACC = 0.78, SEN = 0.88, and SPE = 0.68) in NA groups (Figure 8A).

Furthermore, we pooled the data from the two *APOE* genotyping groups and tracked clinical symptoms using SVR model. The results revealed that even though the aggregated analysis included the largest number of participants, individual-specific functional connectivity in this cross-genotype cohort were unable to estimate MMSE symptom above the chance level ($r = 0.18$, $p = 0.139$, Figure 8B). Similar results were also obtained in LIMM symptom estimation. Individual-specific functional connectivity failed to yield LIMM estimation which was not significantly correlated with the observed LIMM symptom rating ($r = 0.23$, $p = 0.108$, Figure 8C).

Our findings demonstrated the dissociative functional connectivity alterations between *APOE* $\epsilon 4$ carriers and *APOE* $\epsilon 4$ noncarriers in all clinical subgroups. We have rewritten this part and provided detailed results in the section of “*Estimations of symptom dimensions perform better within different APOE genotyping groups*” (Line 1-12, Page 16).

Figure 8. Prediction of clinical groups and symptoms across *APOE* genotyping groups. (A) The AUCs for *APOE* $\epsilon 4$ carriers versus *APOE* $\epsilon 4$ noncarriers for the 5-fold cross-validation in NA, MCI, and AD groups, respectively. (B) Functional connectivity was not able to estimate the MMSE scores in the elderly participants across two *APOE* genotypes. (C) Functional connectivity was not able to estimate the LIMM scores in the cross-genetic cohort.

Minor Comments

Comment 1: “Panel labels (A, B, C, ...) in Figures 4-8 are missing.”

Response: We appreciate the Reviewer for pointing out that. We have now added these panel labels in the updated Figures 4-8.